# The Roles of Neutrophils in Cytokine Storms

**DOI:** 10.3390/v13112318

**Published:** 2021-11-21

**Authors:** Lily Chan, Negar Karimi, Solmaz Morovati, Kasra Alizadeh, Julia E. Kakish, Sierra Vanderkamp, Fatemeh Fazel, Christina Napoleoni, Kimia Alizadeh, Yeganeh Mehrani, Jessica A. Minott, Byram W. Bridle, Khalil Karimi

**Affiliations:** 1Department of Pathobiology, Ontario Veterinary College, University of Guelph, Guelph, ON N1G 2W1, Canada; lchan12@uoguelph.ca (L.C.); jkakish@uoguelph.ca (J.E.K.); vanderka@uoguelph.ca (S.V.); ffazel@uoguelph.ca (F.F.); napoleoc@uoguelph.ca (C.N.); ymehrani@uoguelph.ca (Y.M.); minott@uoguelph.ca (J.A.M.); 2Department of Clinical Sciences, School of Veterinary Medicine, Ferdowsi University of Mashhad, Mashhad 91779-4897, Iran; n.karimi@mail.um.ac.ir; 3Division of Biotechnology, Department of Pathobiology, School of Veterinary Medicine, Shiraz University, Shiraz 71557-13876, Iran; s.morovati@shirazu.ac.ir; 4Department of Pharmaceutical Sciences, College of Pharmacy, University of Illinois at Chicago, Chicago, IL 60612, USA; kaliza2@uic.edu; 5Department of Diagnostic Medicine & Pathobiology, College of Veterinary Medicine, Kansas State University, Manhattan, KS 66506, USA; kimia@vet.k-state.edu

**Keywords:** cytokine storm syndrome, neutrophil, cytokine, innate immunity, NETosis

## Abstract

A cytokine storm is an abnormal discharge of soluble mediators following an inappropriate inflammatory response that leads to immunopathological events. Cytokine storms can occur after severe infections as well as in non-infectious situations where inflammatory cytokine responses are initiated, then exaggerated, but fail to return to homeostasis. Neutrophils, macrophages, mast cells, and natural killer cells are among the innate leukocytes that contribute to the pathogenesis of cytokine storms. Neutrophils participate as mediators of inflammation and have roles in promoting homeostatic conditions following pathological inflammation. This review highlights the advances in understanding the mechanisms governing neutrophilic inflammation against viral and bacterial pathogens, in cancers, and in autoimmune diseases, and how neutrophils could influence the development of cytokine storm syndromes. Evidence for the destructive potential of neutrophils in their capacity to contribute to the onset of cytokine storm syndromes is presented across a multitude of clinical scenarios. Further, a variety of potential therapeutic strategies that target neutrophils are discussed in the context of suppressing multiple inflammatory conditions.

## 1. Introduction

The concept of a cytokine storm was first identified in acute graft-versus-host disease following the transplant of hematopoietic stem cells [1]. Cytokine storm syndrome (CSS) is used as a general term entailing diverse immune-related dysregulations that are triggered by distinct driving factors and result in systemic inflammation, clinical symptoms, and sometimes multiple organ dysfunction [2]. The signs and duration of CSS can differ based on the cause and treatments [3]. The clinical presentations of CSS could be attributed to the activation of complement, kinins, clotting, and fibrinolysis cascades coupled with high concentrations of cytokines [4]. Various cytokines are found to increase in this syndrome, the most common being interleukin (IL)-6, IL-10, interferon (IFN)-γ, monocyte chemoattractant protein (MCP)-1, and granulocyte-macrophage colony-stimulating factor (GM-CSF) [5]. In addition, tumor necrosis factor (TNF), IL-1, IL-2, IL-2-receptor-α, and IL-8 have also been implicated in CSS [6,7]. The levels of C-reactive protein (CRP) and ferritin, as non-specific markers of inflammation, may also rise [8,9].

Overall, inflammation consists of a series of biological mechanisms which contribute to the activation of both innate and adaptive immunity. Cytokines are key components of inflammation and immunoregulation and have short half-lives. Moreover, the production of cytokines is commonly controlled by anti-inflammatory mechanisms. As a result, these molecules do not usually modulate inflammation much beyond the site of infection and/or damage [2]. Uncontrolled concentrations of these inflammatory factors, especially beyond the site of danger, can be accompanied by systemic complications. In other words, there are disadvantages to an excessive immune response such as CSS, which can result in damages that outweigh the benefits.

Cytokine storms are often driven by the unrestrained activation of excessive numbers of leukocytes, including first-responder neutrophils that can cause collateral damage to inflamed blood vessels and other tissues that they infiltrate [10]. Exposure to high doses of pathogens, especially when these pathogens get distributed systemically, such as during sepsis, is a potent trigger of CSS [11]. The overproduction of cytokines can also be caused by the prolonged activation of leukocytes, for instance, what occurs with macrophage activation syndrome [12]. The dysfunctional resolution of inflammatory reactions, such as what occurs with primary lymphohistiocytosis syndrome, is yet another cause of CSS [13]. The innate leukocytes, namely neutrophils, monocytes, macrophages, dendritic cells, and natural killer (NK) cells are among the first line of defense against pathogens and tissue damage and are known to be involved in cytokine secretion and inflammatory reactions [14]. Some of the many roles of innate leukocytes in the development of CSS include the abundant production of cytokines by activated macrophages [15], the diminished cytolytic effect of NK cells leading to longer antigenic stimulation of B and T cells [13], along with the secretion of cytokines and formation of neutrophil extracellular traps (NETs) by neutrophils [16,17,18]. Although multiple cell types in the innate arm of the immune system can contribute to cytokine storms, this review will focus on the contribution of neutrophils in the context of viral and bacterial infections, cancers, and autoimmune disorders. An overview of the potential pharmacologic interventions against CSS that target neutrophils through distinct mechanisms will also be discussed.

## 2. Immunobiology of Neutrophils and Neutrophil-Derived Cytokines

Neutrophils are part of the cellular components that make up the innate immune system and are usually the first cells to arrive at sites of infection, inflammation, and injury via chemotaxis in the blood, followed by extravasation into tissues [19,20]. They develop from precursors in the bone marrow and are released into circulation. They circulate in the body, ready to detect agents that will attract and activate them, including pathogen-associated molecular patterns (PAMPs), damage-associated molecular patterns (DAMPs), or specific cytokines secreted by leukocytes [21]. Once detection and activation occur, neutrophils will initiate an immune response that involves the recruitment and activation of other leukocytes, as well as sending signals to promote the proliferation and development of neutrophils in the bone marrow to assist in the response against a pathogen or insult [22]. Neutrophils normally use three strategies to combat pathogens and prevent their dissemination throughout the body. These include phagocytosis, NETosis, and degranulation [19,20,21]. Phagocytosis involves the engulfment, internalization, and intracellular degradation of pathogens. The uptake of pathogens into neutrophils is sometimes facilitated by antibodies that can be bound to neutrophils in a non-specific fashion via receptors that recognize the constant fragment of immunoglobulins. This process is known as opsonization. Being very small cells, neutrophils have a limited phagocytic potential and die shortly after the internalization of pathogens. In NETosis, the pathogens that are too large to be phagocytized are instead immobilized by the neutrophils, which externalize a web-like group of fibers derived from DNA that are embedded with histones and other antimicrobial molecules [21]. NETosis can contribute to the inflammatory responses that drive autoimmune diseases such as psoriasis and rheumatoid arthritis [23]. Neutrophils usually undergo cell death via apoptosis after releasing the traps and clearing the infection [19]. However, some neutrophils fail to undergo apoptosis after NETosis. This is known as vital NETosis, which is rapid, and the cells remain active and functional, thereby contributing to hyperinflammatory reactions and immunopathogenesis [24]. The ability of neutrophils to play roles in multiple physiological and pathophysiological processes, such as hematopoiesis, autoimmune disorders, inflammation, and angiogenesis, also results from their capacity to secrete diverse cytokines [25]. The pro-inflammatory cytokines IL-1α, IL-1β, IL-6, IL-16, IL-18, and macrophage migration inhibitory factor (MIF), along with enzymes released from neutrophil granules (including neutrophil elastase, myeloperoxidase, azurocidin, enolase-1, and lysozyme) [26], are believed to contribute to the immunopathogenesis of CSS [20]. The crosstalk neutrophils partake in with other leukocytes could potentially amplify immune responses, thereby promoting CSS. Of concern would be the relationship between macrophages and neutrophils, since they function in similar manners. Generally, they can work together to protect the host and would typically be considered a powerful protective duo. However, when their communications are not regulated properly, this can result in a propagation of pro-inflammatory cytokine signaling. This happens because they can feed off each other and create an auto-amplifying immune response [27,28]. Communications between neutrophils and macrophages also determine the outcome of tissue repair, which involves efferocytosis (dying neutrophils being rapidly engulfed by macrophages) and reverse migration (neutrophils migrating out of the tissue) [29].

Unique subsets of neutrophils displaying distinct morphologies and functions have been discovered. These neutrophil subpopulations can alter their phenotypes to adapt to different physiological stressors and conditions [19]. Neutrophil subsets include tumor-associated neutrophils (TANs), low-density neutrophils (LDNs), normal-density neutrophils (NDNs), and myeloid-derived suppressor cells. Additionally, neutrophils with different functions were identified by a collection of cell surface markers. These subsets commonly have distinct functions, depending on the stressor or stimulus [30]. For instance, neutrophils with a CD62L^low^CD16^high^ phenotype enhance T cell activation in the allergic response and, contrarily, suppress the immune responses in viral infections [30]. Another example includes CD177^+^ neutrophils, which play a role in transmigration and exit from circulation to travel towards sites of infection to perform their functions [30]. Some subsets, such as B cell-helper neutrophils (NB_H_ cells) [31], are capable of interacting and communicating with other cells involved in the immune response, including B cells, by promoting antibody production [30]. Therefore, neutrophils are also involved in adaptive immunity through their ability to signal and support B lymphocytes, T lymphocytes, and antigen-presenting cells, using synthesized cytokines [21]. Furthermore, some neutrophil subsets are important in resolving immune responses, reducing inflammation, and restoring homeostatic conditions [19]. Macrophages can be activated by engulfing apoptotic neutrophils to stimulate the anti-inflammatory processes that terminate the immune response once an infection is cleared [21]. Other neutrophils will exit the body as a component in pus, or return to circulation via transmigration [19]. Therefore, disturbances in the differentiation or function of neutrophils can hinder the resolution of inflammation and lead to uncontrolled inflammation and CSS.

## 3. Neutrophil Hyperactivation and CS

### 3.1. Viral Infections

The term cytokine storm in the context of viral infections has been described for a limited number of viruses, including the influenza virus [32], respiratory syncytial virus (RSV), cytomegalovirus (CMV) [33], variola virus [34], Epstein-Barr virus (EBV) [35], severe acute respiratory syndrome coronavirus (SARS-CoV) [36], middle east respiratory syndrome coronavirus (MERS) [37], and SARS-CoV-2 [37]. Adaptive immunity has been implicated as the major responder to viral invasion [38] and less attention has been paid to the role of innate immunity, especially neutrophils, during viral infections. Influenza virus-mediated dose-dependent neutrophil infiltration into lungs [39], RSV-mediated extension of neutrophil life span [40], rapid neutrophil recruitment into the central nervous system (CNS) in West Nile virus (WNV) infections [41], dissemination of WNV [42] or CMV [43] throughout the body by using neutrophils as Trojan horses, and the replication of EBV in these cells [44] are some common examples of the active roles neutrophils can take during viral infections. Neutrophilic immunity can always act as a double-edged sword. These granulocytes can potentially inflict serious tissue damage, leading to organ failure, through the induction of strong inflammatory responses during viral clearance.

#### 3.1.1. RSV

Neutrophils are the most abundant leukocytes in the lungs and airways of patients infected with RSV [45]. Some researchers reported that neutrophils are present in 93% of upper and 76% of lower airway cellular infiltrates [46,47]. Although the earliest stages of neutrophil recruitment are initiated by DAMPs [48], CD8^+^ T lymphocytes were observed to induce the influx of neutrophils during RSV infections in mice [49]. Neutrophils are attracted to the site of infection by chemokines such as IL-8, CXCL1, and CXCL2 [50], and can cause epithelial damage. IL-8 is the most potent neutrophil chemoattractant, which is released by the epithelial cells lining the airways [46]. Mouse studies clearly demonstrate that, while the chemokines MIP-2 and KC act via CXCR2 on neutrophils to mobilize and recruit the cells to the site of inflammation, G-CSF down-regulates the expression of CXCR4 on neutrophils, thus reducing their retention in the bone marrow and mobilizing neutrophils into the circulation [51,52].

It has been reported that RSV can interfere with apoptosis, thereby delaying the death of neutrophils [53] by affecting the nuclear factor-κB- and phosphoinositide 3-dependent pathways [40]. However, the role of PRRs in RSV-mediated extension of neutrophil survival has not been thoroughly addressed. It is assumed that the viral protein-TLR4 engagement and activation of some transcription factors leads to the production of anti-apoptotic cytokines, such as IL-6, from neutrophils, thereby increasing their longevity [40,45,46,47]. This mechanism can exacerbate the immunopathological effects of neutrophils in the airways.

During infection with RSV, neutrophils enter and are retained within the infected respiratory epithelium. This is mediated by the adhesion molecules CD11b, CD18 (intercellular adhesion molecule [ICAM]-1), and CD54, which are highly expressed on the surface of neutrophils. This is complemented by the expression of neutrophil platelet endothelial cell adhesion molecule (PECAM)-1 (also known as CD31) and endothelial PECAM-1 on the endothelial cells. These interactions allow neutrophils to extravasate from blood vessels into the infected tissue [54]. Neutrophil-epithelial cell attachment can be further enhanced by the expression of the adhesion molecules ICAM-1, E-cadherin, CD9, CD44, vascular cell adhesion molecule (VCAM)-1, and major histocompatibility complex (MHC) class I and class II on airway epithelial cells [55,56]; and CD11b, CD18, CD54, Mac-1, and lymphocyte function-associated antigen (LFA)-1 (also known as CD11a/CD18) on neutrophils [56,57]. Moreover, RSV increases alveolocapillary permeability [58]. These events lead to neutrophil infiltration and subsequent epithelial detachment. It is noteworthy that not only active RSV, but also the inactivated virus, can promote neutrophil-epithelial cell interactions [56]. This fact is of value in RSV vaccine development. When infiltrating the epithelial lining of airways, neutrophils can kill RSV-infected target cells through complement-dependent cytotoxicity.

Furthermore, RSV induces NETosis, as the attachment of the virus fusion protein to neutrophil-expressed TLR4 incites the massive production of NETs in ERK and p38 mitogen-activated protein kinase (MAPK) phosphorylation-dependent manners. The NETs can potentially fill airspace in the lungs and impair their functions [59]. Moreover, neutrophils can damage healthy areas of the respiratory system on account of bystander activation [60]. Finally, the resolution of inflammation is accomplished by the induction of apoptosis in the infiltrating neutrophils. Macrophages detect and phagocytose the apoptotic neutrophils, thereby shutting down inflammation [61]. Therefore, these observations suggest that excessive NETosis is likely a contributing cause of CSS and tissue damage from RSV infection.

#### 3.1.2. Influenza Virus

Immune responses against influenza viruses start with sensing the virus inside infected airway epithelial cells through PRRs, namely TLRs, nod-like receptors, or retinoic acid-inducible gene (RIG)-I-like receptors. Triggering the signaling pathways of these PRRs results in the production of antiviral proteins, such as IFNs, that stimulate uninfected cells to shift to antiviral states to block viral replication [62,63,64]. Influenza virus-infected epithelial cells produce pro-inflammatory cytokines and chemokines such as IL-1β, TNFα, IL-6, CCL5, CCL2, CXCL8, and CXCL10 [65]. However, these cytokines can have detrimental effects and the potential to induce a cytokine storm if they exceed the necessary concentration needed to mediate viral clearance. Indeed, overly robust cytokine secretion and inflammatory responses are the main factors associated with fatal cases of infection with the H5N1 influenza virus [66].

Neutrophils have a complex role during an infection with influenza A viruses (IAVs). The accumulation of neutrophils in the airway epithelium after infection with an IAV largely depends on the dose and strain of the virus [67,68]. Neutrophil recruitment is mediated by several chemokines, including CCL3, CXCL-1, and CXCL2. In the lungs, the secretion of matrix metalloproteinase (MMP)-9 enables neutrophils to disrupt the integrity of the endothelial basement membrane to facilitate their entry via CXCR2-mediated chemotaxis [69]. Phagocytosis, cytokine release, and the secretion of antimicrobial peptides are considered the main roles of neutrophils in combating influenza virus infections. Cathelicidin, as a cationic antimicrobial peptide released by neutrophils, showed a promising role in the inhibition of viral replication and pro-inflammatory mediator secretion [70,71].

Seasonal and pandemic influenza strains have been observed to have different effects on human neutrophils. For example, neutrophils challenged with seasonal IAV showed enhanced bacterial clearance, while pandemic strains resulted in defective neutrophil functions and, consequently, an increased risk of secondary bacterial infection in patients infected with IAV [72]. Neutrophil NETosis can prevent viral propagation and is initiated by the secretion of TNF-α and cathelicidins in the lungs during an influenza infection [69,73]. It has been reported that the NET formation strategy differs depending on the influenza strain. For example, seasonal strains are probably more inhibited by NETosis, in comparison with the pandemic strain H1N1 [74]. On the other hand, excessive formation of NETs can also exacerbate disease severity through the bystander destruction of pulmonary endothelium and epithelium leading to acute lung injury [75,76].

It has been revealed that neutropenia is associated with a higher viral load in the lungs, as well as higher mortality in mice infected with IAV [77]. Although the protective role of neutrophils in influenza infections is clearly defined, an excessive release of cytotoxic reactive oxygen species (ROS) leading to lung damage can result from the dysregulated accumulation of neutrophils, especially in the context of infections with highly pathogenic H5N1 and 1918 pandemic IAVs [78]. Neutrophils also have the potential to act as antigen-presenting cells, thereby activating anti-viral CD8^+^ T cells within infected lungs [79].

During an infection with IAV, activated transcription factors, such as NF-κB and interferon regulatory factor (IRF)-3/7, work in a synergistic way to boost pro-inflammatory responses. The balance between pro-inflammatory and anti-inflammatory responses defines the outcome of IAV infections [80,81,82]. Recently, it was shown that glucose metabolism is a driving force underlying the development of cytokine storms during some infections with IAV. It has been determined that O-linked β-N-acetylglucosamine (O-GlcNAc) transferase (OGT)-mediated O-GlcNAcylation plays a pivotal role in IAV-induced cytokine storms. Moreover, IRF5 is critical for the induction of cytokine storms in this context [83].

#### 3.1.3. Highly Pathogenic Coronaviruses

The pandemic caused by the outbreak of infections with SARS-CoV-2, which is the causative agent of the coronavirus disease that emerged in 2019 (COVID-19), has resulted in an increased focus in the scientific community on overly robust and toxic immune responses against highly pathogenic coronaviruses. Neutrophils are the drivers of virally-mediated cytokine storms induced by highly virulent human coronaviruses (i.e., SARS-CoV, MERS-CoV, and SARS-CoV-2) [37]. This is exacerbated by the fact that these viruses can directly trigger an apoptotic pathway in the epithelial cells of the lungs [84].

After alveolar infection with SARS-CoV-2, neutrophils are recruited to the infected alveoli by CXC chemokines and IL-6 [85,86,87,88,89,90]. Complement components, particularly C5a and C3a, are other crucial players of innate immunity that are related to the exaggerated recruitment of neutrophils during the progression of COVID-19 [91,92,93]. A similar pro-inflammatory cytokine profile and complement activation were observed in patients infected with SARS and MERS [94,95,96]. Neutrophilia contributes to poor prognosis and high mortality in severely ill patients with SARS, MERS, and SARS-CoV-2 infections [96,97,98,99,100,101]. The neutrophil-to-lymphocyte ratio (NLR) has been noted as a hallmark of COVID-19 severity [97,98,102,103,104]. Specifically, the high neutrophil to CD4^+^ lymphocyte ratio (NCD4LR) corresponds with a delayed viral clearance [105]. Furthermore, the neutrophil to albumin ratio is another factor for predicting the mortality of patients with COVID-19 [106]. Meizlish et al. [107] proposed a unique neutrophil activation signature, including neutrophil activators (IL-8 and G-CSF) and effectors (resistin, lipocalin-2, and hepatocyte growth factor), which is linked to the absolute neutrophil count and disease intensity.

High neutrophil counts in the severe form of COVID-19 are accompanied by the production of immature CD10^+^ neutrophils and dysfunctional mature neutrophil clusters [108,109]. Neutrophils express a wide-ranging repertoire of PRRs and are able to recognize viral PAMPs, subsequently becoming activated to promote the killing of the virally infected cells via multiple mechanisms, including NETosis [108,110,111]. However, mouse studies demonstrated that the effector functions of neutrophils do not limit viral replication, which is critical for a virus’s ability to infect its host and spread [112]. Additionally, the formation of NETs contributes to the development of acute lung injury, as demonstrated in patients that are critically ill with COVID-19. The augmented secretion of serine proteases and neutrophil elastase, along with the consumption of protease inhibitors during NETosis, can lead to acute respiratory distress syndrome (ARDS), as observed in some cases of infections with SARS-CoV or influenza viruses [108,113]. In this respect, a positive relationship was found between ELANE (a neutrophil elastase) expression and neutrophilia in patients with SARS [114]. The ELANE-mediated neutrophilia also correlates with pulmonary hemorrhagic lesions in patients with SARS. Of note, ELANE upregulation is associated with the elevated expression of CXCL1, which acts as a neutrophil chemoattractant and is also released from activated neutrophils in the lungs [114]. Augmented concentrations of circulating NETs are not only related to lung injury but also cause vascular thrombosis in the liver, heart, and kidney [17,115,116]. Furthermore, the release of a large amount of ROS from neutrophils and the activation of the NF-κB pathway exacerbate the CSS and cause thrombosis following infection with SARS-CoV-2 [117]. This situation increases hypoxia and can lead to multiple organ failure.

### 3.2. Bacterial Infection

Inflammatory reactions occur during bacterial infections [118], and neutrophils are recruited to the site of inflammation by pro-inflammatory cytokines, such as IL-8 [118,119]. Neutrophils identify pathogen PAMPs on bacteria [120], including surface lipopolysaccharides or peptidoglycans that usually activate TLRs or other PRRs [121,122]. Consequently, the neutrophilic elimination of bacteria through phagocytosis, degranulation, or NETosis is initiated, and neutrophils will secrete a variety of cytokines [123]. The infection could become deleterious if the neutrophils fail to eliminate pathogens. This can occur when bacteria develop mechanisms to evade neutrophils [124]. For example, *Staphylococcus aureus* biofilms can evade host immune responses by masking the detection of PAMPs. *S. aureus* biofilms have been shown to produce leukocidins, such as Panton-Valentine leukocidin and *γ*-hemolysin AB, which cause neutrophils to form NETs [125]. The antimicrobial activity of *S. aureus* can persist because NETs are often ineffective in clearing biofilms [125]. As a result, some changes can occur in the inflammatory response to bacteria, leading to a loss of homeostasis, which can potentiate sepsis [11]. Sepsis [126] is a severe clinical syndrome that results from the systemic distribution of bacteria via the blood, and elicits an activation cascade that can lead to dysregulated immune responses causing auto-amplifying cytokine production and CSS [11].

Weber et al. showed, in a mouse model, that IL-3 exacerbates inflammation in sepsis [127]. IL-3 is influential in the production, proliferation, and survival of leukocytes, including neutrophils. They demonstrated that IL-3 was indirectly responsible for generating cytokine storms by creating a large population of cells that produced high quantities of cytokines once bacterial components were detected [127]. Bhatty et al. investigated the effects of alcohol on leukocyte proliferation and the occurrence of cytokine storms in a mouse model. The results suggested that the consumption of high amounts of alcohol at the late stages of sepsis caused a rapid rise in the number of viable bacteria and cytokine levels [128]. Therefore, a cytokine storm following sepsis is considered one of the main examples of CSS due to bacterial infections. An overwhelming pathogen load, resulting in extensive degranulation and NETosis, in addition to the imbalanced and long-lasting inflammatory response because of the immune-evading capacity of some bacteria, could be suggested as the leading cause of CSS following sepsis. There is limited research on the pathophysiology of cytokine storms caused by bacterial infections, despite these being severe clinical scenarios. Further research should be completed to investigate how neutrophils act to clear bacterial infections, and what consequences there can be if the immune response is dysregulated.

### 3.3. Cancers

Unresolved tissue inflammation is a hallmark of the tumor microenvironment. This chronic inflammation arises as a result of either the persistence of carcinogens or a failure of the resolution mechanisms which control the inflammatory process [129]. Chronic inflammation promotes tumorigenesis by dampening immune responses or enhancing DNA damage through the overproduction of ROS [128,129]. As described in many studies, inflammatory bowel disease, non-alcoholic fatty liver disease, chronic hepatitis, and numerous other diseases associated with chronic inflammation increase the risk of cancer development [130,131,132]. Reciprocally, the loss of p53 inducing the secretion of Wnt ligands and ROS promotes systemic inflammation and creates a constant state of cellular distress [129,133]. Neutrophils are often the first cells to arrive at the tumor microenvironment during the initiation phase of carcinogenesis, and their presence is often indicative of the prognosis. The release of IL-8 by cancerous cells attracts neutrophils to the local area [134,135,136]. Elevated serum levels of IL-8 correlate with a worse prognosis in advanced cancers [137]. The NLR in blood is commonly augmented in patients with cancers, and high levels of intra-tumoral neutrophils are associated with unfavorable survival [134].

Cancer cells can redefine the immunological landscape by releasing cytokines and chemokines. In a study conducted by Nishida et al. [138], the expression of inflammation-related genes in cancer cells promoted neutrophil-dependent lung metastasis in clear cell renal cell carcinoma through epigenetic remodeling. These cytokines and chemokines also disrupt the neutrophil retention and release balance in the bone marrow. Indeed, the release of adenosine triphosphate, G-CSF, IL-8, CXC, and leukotriene B4, as well as other DAMPs and chemokines from cells within the tumor microenvironment, promotes neutrophil localization in the tumor [136,139,140,141].

There are two different subsets of tumor associated neutrophils (TANs) that exist in the tumor microenvironment [142]. They are identified as N1 and N2 neutrophils, which possess anti-tumorigenic and pro-tumorigenic phenotypes during the early and late stages of cancer progression, respectively [143]. Indeed, cancer cells and leukocytes within the tumor microenvironment produce chemokines, such as tumor growth factor (TGF)-β, that support the development of N2 neutrophils [142]. Tumor cells also secrete pro-survival signals allowing the TANs to survive in the environment for longer than typical neutrophils [144,145].

The N1 neutrophils with anti-tumorigenic characteristics release pro-inflammatory and immunostimulatory cytokines, while N2 neutrophils promote cancer progression. The N2 neutrophils support tumor angiogenesis, invasion, and metastasis following the release of ROS, oncostatin M, MMP-9, and neutrophil elastase [146]. Moreover, N2 neutrophils further suppress the immune system by secreting CCL2 and CCL17 and, subsequently, recruiting regulatory T cells to sites of tumorigenesis [147,148]. The presence of TGF-β favors the infiltration of N2 neutrophils with high expression levels of CXCR4, vascular endothelial growth factor (VEGF)-A, and MMP-9 [142]. The absence of TGF-β, coupled with the presence of type 1 IFNs, promotes N1 neutrophil maturation while suppressing neutrophil migration to the tumor tissue [144,149,150].

Cancer cells can release ATP as a signal to guide neutrophils to the tumor [151]. Tumor-derived CD73 catalyzes the hydrolysis of ATP to adenosine, which further promotes tumorigenesis [152]. In addition to immunosuppression, adenosine contributes to the formation of NETs through the activation of A1 and A3 neutrophil receptors [149]. The NET DNA acts as a chemotactic factor, attracting circulating tumor cells, and sequestering them at distant sites, promoting metastasis [150]. Neutrophils have been implicated in the development of premetastatic niches through NETs in many cancers, including ovarian cancers [153]. Furthermore, NETs have been shown to activate dormant cancer cells during inflammation through the accumulation of neutrophil proteases [154].

T-cell-engaging immunotherapeutic agents, such as chimeric antigen receptor (CAR) T-cell therapy, have shown impressive therapeutic activity in several hematologic malignancies [155]. However, cytokine release syndrome (CRS), a systemic inflammatory response, represents the most frequent adverse effect of this therapy [155]. Jain et al. demonstrated that high-grade CRS was significantly associated with the absence of complete hemoglobin, platelet, neutrophil, and white blood cell count recovery at one-month post-treatment [156]. The hematologic toxicity associated with CAR T-cell therapy is often attributed to the lymphodepleting chemotherapy regimen used prior to the therapy [155]. Dunleavy et al. hypothesized that perturbation in stromal derived factor-1, a chemokine responsible for regulating hematopoietic stem cell migration, following B-cell recovery stunts neutrophil release from the bone marrow [157]. However, Jain et al. demonstrated that the frequency of cytopenia cannot be fully explained by these prior therapies, and likely includes CAR T-cell-related mechanisms [156]. They observed that the cytokines commonly associated with CRS were not significantly elevated in the patients without a complete blood count recovery at one month. However, the concentration of macrophage-derived chemokine and other growth factors were decreased [156]. Further work must be done to fully evaluate the effect of the innate inflammatory response, including the impact of neutrophils, on the development of CRS as related to these immunotherapeutic agents.

### 3.4. Autoimmunity

Neutrophils are the most abundant type of granulocyte in the immune system [158]. The improper activation of these cells by cytokines, chemokines, and autoantibodies may misdirect their effector properties in ways that can have destructive effects on host tissues, leading to autoimmune diseases [159]. The exact role of neutrophils in autoimmune disorders has not been well defined. However, accumulating evidence indicates the significant contribution of neutrophils to the pathogenesis of autoimmune diseases through presentation of antigens and modulation of the function of other cells. Recent studies suggest neutrophils play a part in several phases of autoimmunity, such as immunization, transition, and effector phases. The secretion of autoantigens when neutrophils are either activated, in the case of autoimmune vasculitis, going through apoptosis, or forming NETs are several examples. Experiments investigating antibody-mediated depletion of neutrophils using animal models have led to discoveries concerning the roles of neutrophils in autoimmune diseases [160].

Autoimmune neutrophils present in high numbers at the site of autoimmune reactions and employ numerous mechanisms to exert their detrimental effects on the host tissues, either directly through some effector molecules, or indirectly by communication with other leukocytes. Some of these impacts include releasing several cytokines and chemokines functioning both in innate and adaptive immunological mechanisms [161], secreting proteases that harm host cells and soluble proteins [162], forming NETs, producing ROS [163,164,165], and bringing about trans-signaling through several receptors, such as the IL-6 receptor [166]. Moreover, in a number of studies, neutrophils were shown to be engaged in the pathogenesis of autoimmune diseases through involvement in a lipid-cytokine-chemokine cascade [167].

#### 3.4.1. Multiple Sclerosis (MS)

MS is an inflammatory disorder that attacks the CNS [168]. The results of an investigation of autoimmune encephalomyelitis (EAE), which is a well-known mouse model for MS, indicated a notable rise in the neutrophil count during the acute phase of the disease, followed by a drop during the remission phase [169]. Moreover, studies indicated a higher NLR in patients with MS compared to healthy controls. Indeed, a high NLR has been considered an independent predictor of disability progression in this disease [170]. Naegele et al. also showed that neutrophils are more numerous in patients with MS compared to healthy people. Moreover, they noted that neutrophils in these patients showed a primed state due to lower apoptosis, increased degranulation, augmented oxidative burst, and elevated serum concentrations of NETs, along with higher expression of TLR-2, fMLP receptor, IL-8 receptor, and CD43 [171]. Therefore, neutrophil-mediated CSS in patients with MS likely originates from the predominantly proinflammatory function of these cells [172].

#### 3.4.2. Rheumatoid Arthritis (RA)

RA is a chronic autoimmune disorder affecting many joints, including those in the hands and feet, causing pain, progressive disability, and early death. Previous studies not only demonstrated a correlation between NLR and disease activity, but also presented the combination of NLR and platelet-to-lymphocyte ratio as a prognostic factor for RA [173,174,175,176]. The major destructive part neutrophils play in RA pathogenesis is due to the autoantibody production against citrullinated peptides during NETosis. Auer et al. showed that mouse and human neutrophils produced either citrullinated NETs, uncitrullinated NETs, or a mixture of both in vitro. The TNF-α released by neutrophils can, in turn, enhance the production of CCL18. In other words, neutrophils are capable of recruiting and activating the T or B cells associated with RA [177]. Furthermore, a number of cytokines that are involved in the delayed apoptosis of neutrophils through the induction of anti-apoptotic signals or the inhibition of pro-apoptotic pathways contribute to the development of RA [178]. Additionally, neutrophils may contribute to the production of autoantigens that drive the autoimmune processes underlying RA [165]. A study by Tan et al. demonstrated that the general pathogenesis of RA is associated with the inflammation and imbalanced immune responses establishing CSS [179]. Confirming whether neutrophils exhibit a proinflammatory function that contributes to CSS in MS requires further investigations.

#### 3.4.3. Systemic Lupus Erythematosus (SLE)

SLE is a systemic autoimmune disorder that is responsible for generating inflammation in connective tissues, including cartilage and blood vessels. This chronic autoimmune disease is brought about by a systemic intolerance to nuclear antigens and deposition of immune complexes in various tissues, which in turn can lead to multiorgan failure. Studies have shown that in patients with SLE, LDNs secrete higher concentrations of pro-inflammatory cytokines, including IFN-α and TNF-α [180]. These cells were also reported to be more abundant in patients with SLE [181], and they had elevated pro-inflammatory activity and synthesized more type I IFNs [182]. Midgley et al. reported that the apoptosis of neutrophils was enhanced in patients with SLE in comparison with healthy individuals. In the latter investigation, the apoptosis of neutrophils was shown to be correlated with disease activity biomarkers such as erythrocyte sedimentation rate and double-stranded DNA concentration [183]. Given that in SLE, neutrophils are more activated and dysregulated [184], one could propose that these cells contribute to the loss of tolerance, which may disturb regulatory mechanisms that exist to prevent CSS.

### 3.5. Neutrophils as a Therapeutic Target

With more details being discovered on the mechanisms of neutrophil activation and its roles in infections, cancers, and autoimmune diseases, an increasing number of potential therapeutic strategies targeting neutrophils are emerging. These therapeutic measures are based on two approaches of enhancing or inhibiting neutrophil function. Promoting neutrophil function could be beneficial in severe bacterial or fungal infection, especially in neutropenic conditions, as well as in anti-cancer immunotherapy [185]. On the other hand, excessive neutrophil activation could contribute to immunopathological events, in which case reducing the number of neutrophils or minimizing their activity would be desired. Considering the role of neutrophils in CSS, targeting these cells can be of value for controlling this syndrome and similar conditions. Therefore, it is important to understand some of the medications which have the potential to restrain the activity of neutrophils and dampen their contribution to CSS.

First, the number of neutrophils could be reduced by suppressing neutrophil production or promoting apoptosis of neutrophils [185]. The former can be achieved by inhibiting the regulators of neutrophil homeostasis, including G-CSF, which can be done directly or indirectly by inhibiting the IL-23/IL-17 axis, since it is a regulator of G-CSF. Targeting this mechanism has been shown to be effective against inflammatory diseases in mice and humans [186,187]. As for inducing neutrophil apoptosis, the anti-apoptotic protein MCL1 could be a target because it is necessary for neutrophil survival, and its transcription is influenced by the cyclin-dependent kinases (CDKs) expressed by neutrophils [185]. Rossi et al. showed that the inhibitors of CDK9 expedite the resolution of neutrophil-mediated inflammation in various models [188]. In another study, the targeted delivery of (R)-roscovitine, another CDK inhibitor, into human neutrophils via polymersomes promoted apoptosis in vitro [189].

Moreover, it is possible to target neutrophil recruitment and chemotaxis. Efforts in this regard have mainly been focused on selectins and integrins, as well as the main neutrophil chemokine receptors CXCR1 and CXCR2, with the most promising results being reported in improving lung function and autoimmune diseases [185].

Numerous reports have been published on targeting pathways involved in neutrophil activation. First, Janus kinase (JAK) inhibitors, such as tofacitinib (JAK1/JAK3 inhibitor) and baricitinib (JAK1/JAK2 inhibitor) have demonstrated therapeutic effects for inflammatory diseases, likely by diminishing neutrophil production [188,190]. Furthermore, the tyrosine kinase SYK, which is important in integrin and receptor signaling in neutrophils [191], and the BTK tyrosine kinase family, which are important in neutrophil activation in sterile inflammation [192], are among other targets.

Finally, another approach is to target NETosis pathways to minimize NETs-associated host damage. Okeke et al. showed that the targeted delivery of sivelestat, an in vitro inhibitor of NETs, via a nanoparticle system effectively reduced NET formation in mice [193]. Chirivi et al. engineered an antibody against citrullinated protein as an anti-inflammatory agent in collagen antibody-induced arthritis in mice. These authors demonstrated that the antibody can also inhibit murine and human NET formation [190]. Last, but certainly not least, is the targeting of neutrophil granule enzymes [185]. For example, neutrophil elastase inhibitors have been tested to treat lung diseases which, despite some discouraging results, have shown promise in certain conditions [185]. Given the growing number of therapeutics targeting neutrophils, further research should aim to identify drug combinations that can be effectively used across an array of cytokine storm disorders.

## 4. Conclusions

While neutrophils and their roles in mitigating bacterial infections have been extensively studied and are well characterized, new research advances highlight the complexity and importance of understanding the diversity of neutrophil biology, including the position these first responders of the innate immune system hold as crucial components in limiting viral infection and replication, mediating tumor pathogenesis, and promoting homeostasis. On one hand, neutrophils employ a multifaceted range of tactics to combat an equally diverse range of pathogens and detrimental clinical manifestations to promote restoration, including but not limited to phagocytosis, the formation of extracellular traps, the production of cytokines such as IFNs, and the modulation of innate lymphoid cells and lymphocytes to drive long-term adaptive responses against a broad spectrum of ailments. On the other hand, in this review we have highlighted the destructive potential of neutrophils (Figure 1) in their capacity to contribute to the induction of auto-immune disorders, augment the progression of cancers, cause massive tissue injuries in viral infections and, particularly, promote the onset of CSS across a multitude of clinical scenarios. 

Despite rapid advances in the field, many unknown aspects of the involvement of neutrophils in the clinical manifestation of CSS remains to be clarified. A gap remains in understanding how plastic changes in neutrophil phenotype and functional status over time during the development of CSS occurs, and what distinct neutrophil subsets are primarily involved in augmenting disease severity. The availability of efficient cell isolation techniques, and the identification of surface markers enabling the tracking of the varied neutrophil subsets and their cytokine production profiles, will prove important in increasing the understanding of how each of these different subpopulations individually contribute to CSS onset and regulate disease outcomes. This will invariably improve the development of treatment modalities that target specific detrimental neutrophil subsets without compromising immunity. Despite the existence of a wide array of treatment measures where neutrophils serve as cellular targets, such as in the administration of neutrophil-neutralizing antibodies, such targeted therapies also simultaneously reduce circulating neutrophils, which have profound roles in diverse protective host responses to various insults. It is, therefore, pertinent to consider alternative mitigation strategies for the early control of the cytokine storm, such as through utilizing more target-specific immunomodulators and cytokine antagonists.

Additionally, future research should involve the identification of the molecular mechanisms controlling cytokine expression profiles in neutrophils in both mouse models and humans. Such studies may lead to the identification of novel, neutrophil-specific transcription factors that could further elucidate how neutrophils, and the cytokines they produce, contribute to CSS and collaborate with other myeloid cells to propagate the CSS phenomena across multiple clinical scenarios. Furthermore, the relationship between macrophages and neutrophils, and the influence of their cytokine productions on CSS, should be considered due to the inherent similarities and close partnership between these two types of leukocytes. How neutrophils and neutrophil-derived cytokines are regulated and function in various diseased microenvironments to influence cell-to-cell communication within the innate and adaptive arms of the immune system will be important considerations when attempting to develop efficacious therapeutic interventions for preventing CSS and tissue damage induced by neutrophil-derived cytotoxic molecules. Normalizing the clinical application of assessing NLRs, alongside the continuous monitoring of neutrophil phenotype, function, and cytokine expression profiles can prove predictive of disease severity and outcomes across a wide range of ailments, and may give a prophylactic advantage to clinicians that can utilize this knowledge to identify patients at risk for severe disease. Given the different manifestations of CSS in a multitude of clinical diseases, it is of paramount importance to further recognize and investigate the nature of the initiation and progression of this systemic inflammatory process, which will ultimately be of help to curbing lethal clinical scenarios.

## Figures and Tables

**Figure 1 viruses-13-02318-f001:**
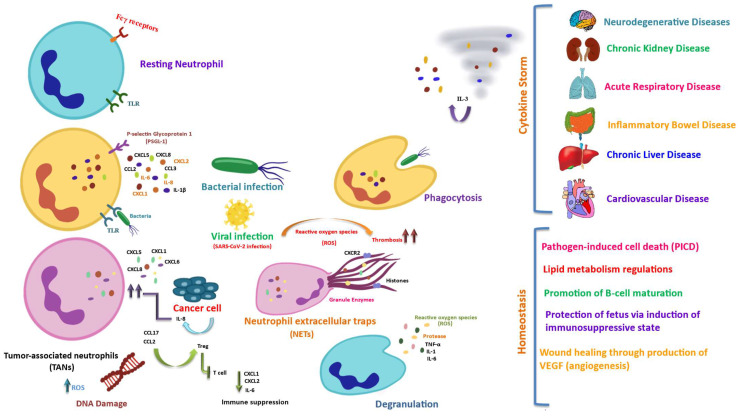
Neutrophils contribute to the pathogenesis of cytokine storm syndrome (CSS). CSS is the discharge of an abnormally high concentration of soluble mediators following the occurrence of an inappropriate inflammatory cytokine response that leads to immunopathological events. It occurs after severe infections, as well as in situations where an inflammatory cytokine response is initiated, exaggerated, but then failed to return to homeostasis. Neutrophils triggered by pathogens (viruses, bacteria, etc.), cancer cells, and autoimmune microenvironmental cues gain the potential to develop CSS.

## Data Availability

Not applicable.

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
