# Peer review of "The Roles of Neutrophils in Cytokine Storms"

_viruses, 2021, doi:10.3390/v13112318_

Round 1

Reviewer 1 Report

This interesting review highlights the advances in understanding mechanisms governing neutrophilic inflammation against viral and bacterial pathogens, in cancers, and in autoimmune diseases, and how neutrophils could influence the development of cytokine storm syndromes. The manuscript is well written and organized. References are quite relevant and updated.

This paper can be very useful for Viruses readers, because it provides very interesting information within the current context of few published studies.

Author Response

We would like to thank the reviewer for the careful assessment of our manuscript. 

Reviewer 2 Report

The manuscript entitled “The Roles of Neutrophils in Cytokine Storms” discussing the effects of neutrophils in various disease involving cytokine storms is intriguing. However, several minor issues remains to be clarified:

  1. What are the mechanisms underlying the development, recruitment and activation of neutrophils?
  2. What is the consequences of the cytokines released by neutrophils in cytokine storms and are there any interaction between cytokines secreted by neutrophils versus those by macrophages?
  3. Figures and/or tables should be provided for the importance of effects of neutrophils in diseases and related mechanisms and potential treatments.
  4. “Virus-infected cells are destroyed through the proteolytic activities of neutrophils, which may or may not involve NETosis.” This sentence is not clear. Do neutrophils specifically recognize and kill viral infected cells? Please provide references.
